# Programmed Death Ligand-1 and Tumor Burden Score Dictate Treatment Responses in Patients with Recurrent or Metastatic Head and Neck Squamous Cell Carcinoma

**DOI:** 10.3390/cancers16091748

**Published:** 2024-04-30

**Authors:** Ming-Yu Lien, Chih-Chun Wang, Tzer-Zen Hwang, Ching-Yun Hsieh, Chuan-Chien Yang, Chien-Chung Wang, Ching-Feng Lien, Yu-Chen Shih, Shyh-An Yeh, Meng-Che Hsieh

**Affiliations:** 1Division of Hematology and Oncology, Department of Internal Medicine, China Medical University Hospital, Taichung 40201, Taiwan; 2School and Medicine, China Medical University, Taichung 40201, Taiwan; 3Department of Otolaryngology, E-Da Hospital, Kaohsiung 82445, Taiwan; 4College of Medicine, I-Shou University, Kaohsiung 82445, Taiwan; 5Department of Otolaryngology, E-Da Cancer Hospital, Kaohsiung 82445, Taiwan; 6Department of Radiation Oncology, E-Da Hospital, Kaohsiung 82445, Taiwan; 7Department of Hematology-Oncology, E-Da Cancer Hospital, Kaohsiung 82445, Taiwan

**Keywords:** tumor burden, survival, cetuximab, pembrolizumab, recurrent or metastatic head and neck squamous cell carcinoma

## Abstract

**Simple Summary:**

Little is known regarding the significance of tumor burden in the treatment of recurrent or metastatic head and neck squamous cell carcinoma (R/M HNSCC). Patients were stratified into high tumor burden (HTB) and low tumor burden (LTB) groups according to their tumor burden score. Our study showed tumor burden was significantly correlated with survival in R/M HNSCC patients, independent of PD-L1 status. HTB patients receiving EPF had better survival than those receiving PPF, regardless of PD-L1 expression. For LTB PD-L1 positive patients, there was a longer survival on PPF than EPF. For LTB PD-L1 negative patients, survival was similar between PPF and EPF. Hence, PD-L1 and TBS should be considered by the multi-disciplinary team for the treatment of R/M HNSCC.

**Abstract:**

Background: The significance of tumor burden for survival is unknown for patients with recurrent or metastatic head and neck squamous cell carcinoma (R/M HNSCC). The purpose of our study was to evaluate the prognostic impact of programmed death ligand-1 (PD-L1) and tumor burden score (TBS) in patients with R/M HNSCC. Patients and Methods: R/M HNSCC patients who were treated with cisplatin, 5-fluorouracil plus cetuximab (EPF) or pembrolizumab (PPF) as first-line treatment were included in our study. PD-L1 and TBS were estimated and correlated with treatment responses. Kaplan–Meier curves were plotted for outcomes estimation. Results: A total of 252 R/M HNSCC patients were included, with 126 high tumor burden (HTB) and 126 low tumor burden (LTB) patients. Median progression-free survival (PFS) was 7.1 months in LTB and 3.9 months in HTB (*p* < 0.001) and median overall survival (OS) was 14.2 months in LTB and 9.2 months in HTB (*p* = 0.001). Patients with LTB had better PFS and OS than those with HTB independent of PD-L1 status. Subgroup analysis showed HTB patients treated with EPF had better survival than those treated with PPF, regardless of PD-L1 expression. For LTB PD-L1 positive patients, there was a longer survival with PPF than EPF, while for LTB PD-L1 negative patients, survival was similar between PPF and EPF. Multivariate analysis exhibited that tumor burden was significantly correlated with OS. Conclusions: Tumor burden is significantly correlated with survival in patients with R/M HNSCC. PD-L1 and TBS should be taken into consideration to determine first-line treatment.

## 1. Introduction

Head and neck caner ranks as the sixth common malignancy, accounting for 5.3% of all cancer cases [1]. It was estimated there were around 890,000 new cases in 2017 by the Global Burden of Disease study [2]. However, the prognosis of R/M HNSCC is miserable with 12–15 months of survival [3]. Current guidelines suggest pembrolizumab with or without chemotherapy as standard first-line treatments for recurrent or metastatic head and neck squamous cell carcinoma (R/M HNSCC) [4]. Pembrolizumab is an immune checkpoint inhibitor against the programmed death-1 (PD-1) receptor. Pembrolizumab was approved for first-line treatment in patients with R/M HNSCC in 2019. Keynote-048 is a phase III study that demonstrated that pembrolizumab alone increased survival significantly in patients with a programmed death ligand-1 (PD-L1) combined positive score (CPS) of 1 or more, while pembrolizumab plus platinum, 5-fluorouracil (PPF) extended survival in the total population [5]. An alternative treatment option is a cetuximab-based regimen. EXTREME is also a phase III study that confirmed that cetuximab plus platinum, 5-fluorouracil (EPF) followed by cetuximab maintenance weekly was superior to chemotherapy alone. The overall response rate (ORR) was increased from 20% to 36%, the median progression-free survival (PFS) was increased from 3.3 to 5.6 months, and the median overall survival (OS) was also increased from 7.4 to 10.1 months [6]. To date, no reliable biomarkers are recommended in decision making about the first-line treatment for patients with R/M HNSCC.

Tumor burden is an emerging factor that might be useful to predict survival in patients with various malignancies treated with systemic treatment. Recently, more and more studies suggested that initial tumor burden exhibited great ability to predict the response to immunotherapy [7]. A retrospective study conducted by Suzuki et al. investigating the impact of tumor burden as well as tumor growth on survival in patients with R/M HNSCC treated with immunotherapy [8]. These results suggest that the tumor growth rate and the sum of baseline tumor lesions are significantly correlated with survival in patients with R/M HNSCC treated with immunotherapy. Of note, the tumor burden score (TBS) was an indicator of tumor burden and was demonstrated as a novel prognostic factor for colon cancer [9], hepatocellular carcinoma [10,11,12] and cholangiocarcinoma [13]. The significance of TBS and PD-L1 for patients with R/M HNSCC is not well-established. Herein, we conducted a multi-institutional retrospective study to investigate the prognostic impact of TBS and PD-L1 in patients with R/M HNSCC treated with EPF or PPF.

## 2. Materials and Methods

### 2.1. Patients

Pathologically confirmed R/M HNSCC patients from 2017 to 2020 at E-Da Hospital, E-Da Cancer Hospital and China Medical University Hospital were retrospectively reviewed. R/M HNSCC patients who were treated with EPF or PPF as first-line treatment for R/M HNSCC were included in our study. The choices of first-line chemotherapy regimen were at the physicians’ discretion. The information of PD-L1 and TBS were retrospectively retrieved. Patients who received cetuximab or immunotherapy before R/M HNSCC were excluded from our study. Patients with tumor recurrence or metastasis within six months after curative chemoradiotherapy were also excluded. Other exclusion criteria included other first-line regimen than PPF and EPF or irregular follow-up intervals. Our study was a retrospective analysis, which was waived from informed consent. Our study was also approved by the Institutional Review Board of E-Da Hospital (EMRP-111-119) and was conducted in accordance with the Declaration of Helsinki. 

### 2.2. Chemotherapy Protocols

For EPF, patients received cisplatin 70–100 mg/m^2^ on day 1 and 5-FU 700–1000 mg/m^2^ on day 1–4 plus cetuximab 400 mg/m^2^ loading on day 1 and then 250 mg/m^2^ weekly on subsequent administration every 4 weeks. For PPF, patients received cisplatin 70–100 mg/m^2^ on day 1 and 5-FU 700–1000 mg/m^2^ on day 1–4 plus pembrolizumab 2 mg/kg or a 200 mg fixed dose every 3 weeks. The dose could be modified according to the underlying disease and side effects. Carboplatin was substituted for cisplatin in patients with poor renal function. Computed tomography was scheduled to evaluate the treatment response periodically.

### 2.3. Programmed Death Ligand-1 and Tumor Burden Score Evaluation

PD-L1 and TBS were obtained by using the archived tissue and last image study before the initiation of EPF or PPF for R/M HNSCC. Programmed death ligand 1 (PD-L1) was presented with tumor proportional score (TPS) and combined positive score (CPS) indicating the percentage of positive membrane staining of tumor cells. TPS and CPS PD-L1 expression was estimated by immunohistochemistry assay using Dako 22C3 PharmDx, while tumor cell (TC) PD-L1 expression was estimated using Dako 28–8 PharmDx IHC assay. Positive PD-L1 expression referred to either TPS > 50%, CPS > 1, or TC > 10%. Negative PD-L1 expression referred to TPS < 50%, CPS < 1 and TC < 10%. Tumor burden was estimated with TBS. TBS was measured by using the following formula: TBS2 = (maximal diameter of largest tumor) 2 + (number of tumor lesions) 2. The cutoff value of TBS in our study was set at 5.66 according to receiver operating characteristic (ROC) curve analysis (Appendix A). Thus, all patients were stratified according to TBS. High tumor burden (HTB) referred to a TBS higher than the cutoff value, while a low tumor burden (LTB) referred to a TBS less than the cutoff value.

### 2.4. Statistical Analysis

All patients’ characteristics were retrospectively retrieved from a medical chart review. The differences between HTB and LTB were compared with chi-square tests. Propensity-score matching was used to diminish patient selection bias. SPSS Version 26 was used for all of the statistical analyses. The caliper value was set at 0.2 and the parameters included gender, age, primary tumor location, initial stage, surgery, chemoradiotherapy, disease status, and first-line chemotherapy. The oncologic outcomes were presented as PFS and OS. PFS was defined as the time from the first day of first-line chemotherapy until tumor progression or final follow-up, while OS was defined as the time from the first day of first-line chemotherapy until the date of death or final follow-up. Kaplan–Meier curves were plotted for survival. Multivariate analysis was also performed with a Cox regression model using “enter” selection to adjust the influences of potential confounders. *p* values were all two-sided and defined to be significant if *p* values < 0.05.

## 3. Results

### 3.1. Patients Characteristics

Initially, there were 315 R/M HNSCC patients enrolled into this study, with 126 HTB patients and 189 LTB patients. After propensity score matching, 252 patients were analyzed for survival prediction, with 126 HTB patients and 126 LTB patients. All clinical and basic characteristics of our patients are presented in Table 1. In brief, 95% were male patients and 61% were younger than 60 years. As for primary tumor locations, the majority were the oral cavity (49%), followed by the oropharynx (26%), hypopharynx (21%), and larynx (4%). P16 status was only available in oropharyngeal cancer patients, accounting for 3.5% P16 positive and 47% P16 negative. Up to 88% of our patients had stage III–IV disease at their first diagnosis. As for treatment history, most patients underwent curative treatment including 63% radical surgery and 79% curative chemoradiotherapy. PD-L1 expression was identified in 73% of our patients with 20% PD-L1 positive and 55% PD-L1 negative. Upon enrollment, 26% of our patients had locally recurrent disease only, while 74% of patients had distant metastasis with or without local recurrence. In terms of first-line treatment, 65% of our patients were treated with the EPF regimen and 35% were treated with the PPF regimen. In the subgroup analysis, patients were classified according to TBS with 126 HTB patients and 126 LTB patients. All baseline data were well balanced between these two groups, including gender, age, primary tumor location, p16 status, initial stage, previous treatment history, PD-L1 status, disease status upon enrollment and first-line chemotherapy regimen.

### 3.2. Survival Outcomes

The median follow-up period of our study was 13 months. At the time of analysis, 86% of the patients were deceased and malignancy was the major reason leading to death. The median PFS of LTB and HTB were 7.1 months versus 3.9 months, respectively (*p* < 0.001), while the median OS of LTB and HTB were 14.2 months versus 9.2 months, respectively (*p* = 0.001). Figure 1 plots the survival curves of PFS and OS. Subgroup analysis was performed with PD-L1 expression status. For patients with negative PD-L1 expression, the median PFS of LTB and HTB were 5.0 months versus 1.6 months, respectively (*p* < 0.001), while the median OS of LTB and HTB were 14.2 months versus 5.0 months, respectively (*p* < 0.001). For patients with positive PD-L1 expression, the median PFS of LTB and HTB were 11.2 months versus 4.8 months, respectively (*p* < 0.001) while the median OS of LTB and HTB were 20.0 months versus 8.5 months, respectively (*p* < 0.001). Figure 2 plots the survival curves of PFS and OS, stratified by PD-L1 status. 

Patients were further stratified according to treatment regimen. Among 126 patients with HTB, 86 patients received EPF and 40 patients received PPF. Among 126 patients with LTB, 77 patients received EPF and 49 patients received PPF. Clinical and basic characteristics of our patients are presented in Table 2. All baseline data were well balanced between EPF and PPF in both HTB and LTB groups including gender, age, primary tumor location, p16 status, initial stage, previous treatment history, PD-L1 status and disease status upon enrollment. For HTB PD-L1 positive patients, the median PFS in EPF and PPF were 5.8 months versus 4.8 months, respectively (*p* = 0.042), while the median OS in EPF and PPF were 9.9 months versus 6.4 months, respectively (*p* = 0.002). For LTB PD-L1 positive patients, the median PFS in EPF and PPF were 10.5 months versus 17.7 months, respectively (*p* = 0.082), while the median OS in EPF and PPF were 12.9 months versus 22.8 months, respectively (*p* = 0.018). For HTB PD-L1 negative patients, the median PFS in EPF and PPF were 6.3 months versus 2.5 months, respectively (*p* = 0.001), while the median OS in EPF and PPF were 9.1 months versus 6.0 months, respectively (*p* = 0.004). For LTB PD-L1 negative patients, the median PFS in EPF and PPF were 11.5 months versus 10.3 months, respectively (*p* = 0.482), while the median OS in EPF and PPF were 18.8 months versus 20.5 months, respectively (*p* = 0.585). 

In summary, HTB patients receiving EPF had better outcomes than those receiving PPF, regardless of PD-L1 expression. For LTB PD-L1 positive patients, there was a longer survival with PPF than EPF, while for LTB PD-L1 negative patients, survival was similar between PPF and EPF. Figure 3 plots the survival curves of PFS and OS, stratified by PD-L1, tumor burden and chemotherapy regimen.

Cox regression analysis was performed with survival for potential prognostic factors and presented in Table 3. Multivariate analysis identified that disease status, PD-L1 expression and tumor burden were independent predictors that correlated with PFS. Moreover, multivariate analysis showed that PD-L1 expression and tumor burden were significant predictors that correlated with OS. 

## 4. Discussion

To our best knowledge, this is the first study to investigate the prognostic impact of PD-L1 and TBS for patients with R/M HNSCC. According to TBS, we easily stratified R/M HNSCC patients into HTB and LTB groups. Our study demonstrated that patients with LTB had significant better survival than those with HTB independent of PD-L1 expression. Interestingly, the optimal first-line chemotherapy regimen seemed different between the HTB and LTB groups. HTB patients treated with EPF had better survival than those treated with PPF, regardless of PD-L1 expression. For LTB PD-L1 positive patients, there was a longer survival with PPF than EPF. For LTB PD-L1 negative patients, survival differences were insignificant between PPF and EPF. Our study confirmed the prognostic role of tumor burden in patients with R/M HNSCC, as well as established the real-world evidence regarding the optimal first-line chemotherapy in patients with HTB and LTB.

Tumor burden simply indicates the total amount of tumor in the body. Accumulating studies have demonstrated that tumor burden is an independent prognostic factor with a negative impact on survival in several types of malignancies, including melanoma [14], lung cancer [15,16,17,18], head and neck cancer [19,20], thyroid cancer [21] and lymphoma [22]. Kim et al. conducted a systematic review and identified the negative impact of tumor burden on baseline immunity and treatment-induced immune responses [7]. More recently, Dall’Olio et al. reviewed various tools for tumor burden evaluation, including computed tomography (CT), 2-deoxy-2-[^18^F]-fluoro-D-glucose positron emission tomography/computed tomography and circulating tumor cell (CTC) and emphasized the poor prognosis among cancer patients with a high tumor burden across all kinds of assessment [23]. 

Our study also demonstrated that R/M HNSCC patients with HTB had poor outcomes as compared with those with LTB in terms of PFS and OS. Suzuki et al. conducted a retrospective study focusing on R/M HNSCC treated with nivolumab in 2020 [8]. They firstly suggested that the sum of the diameter of baseline tumor lesions and tumor growth rate were independently associated with OS and PFS in patients with R/M HNSCC. However, it was hard to realize the tumor growth rate at the beginning of treatment and it was also difficult to calculate the size of all tumors. Based on these reasons, Gr and SumTLs are not widely used in clinical practice. Matoba et al. also conducted another retrospective analysis of 94 patients with R/M HNSCC treated with immunotherapy [20]. They estimated the tumor burden using the number of tumor lesions and the size of the largest tumor lesions. Recently, a novel biomarker of TBS was proposed to define tumor burden. Our study was the first to confirm the prognostication of TBS in R/M HNSCC patients, as well as its predictive role in first-line treatment selection. Further prospective studies with large cohorts are warranted to confirm our conclusions.

The mechanism of how tumor burden could influence survival is uncertain. Previous studies showed cancer cells secreted vascular endothelial growth factor and antitumor cytokines, which inhibits immunological cytotoxicity of T cells [24]. Thus, it can be expected that a larger tumor burden strongly inhibits the antitumor activity of immune cells [25]. Another explanation from preclinical data disclosed that large tumors are more immunosuppressive in comparison with small tumors whether they are primary tumors or metastatic tumors. This immunosuppressive microenvironment directly diminishes the function of the immune system to elicit immune responses against cancer cells [7]. Moreover, larger tumors exhibited greater local and systemic alterations of the immune system, and harbored a more immunosuppressive tumor microenvironment with more immunosuppressive cells and molecules that dampened antitumor activity [26]. Basic research concluded that an increased tumor burden is correlated with increased CD8+ T cell exhaustion, which may reduce the therapeutic efficacy of PD-1 inhibitors [27]. Additionally, some studies have indicated that CD8+ tumor infiltrating lymphocytes (TILs) are impaired with a higher tumor burden as well as showing negative responses to PD-1/PD-L1 inhibitor monotherapy [28]. Huang et al. showed that the reason for clinical failure was owing to an imbalance between tumor burden and reinvigoration of exhausted T cells [29]. In addition, a higher tumor burden is usually associated with reinvigorated Ki67+ CD8 T cells before treatment, which indicates a poor prognosis. Another study also found that the tumor burden in patients with longer survival was usually smaller than those with shorter survival [29]. These results suggested that immune checkpoints inhibitors would be more effective for patients with a lower tumor burden. Our results are consistent with these studies. For patients with LTB, the median PFS was longer in patients receiving PPF than those receiving EPF while the difference in the median OS was insignificant between the two arms. For patients with HTB, the median PFS and OS were superior in patients receiving EPF than those receiving PPF. Further prospective studies with large cohorts are warranted to confirm our conclusions.

There are several inevitable biases in our study. First, only patients with R/M HNSCC treated with either cetuximab or immunotherapy-based treatment were enrolled for analysis. Those R/M HNSCC patients without any chemotherapy for their metastatic disease were excluded. The choice of cetuximab or immunotherapy-based treatment was at the physician’s discretions, rather than randomly controlled. This is a major bias in this study. Second, the measurement of largest diameter was variable. Some tumors were measured in a coronal view, while others were measured in an axial view. Different methods might influence the final diameter of the largest tumor. However, we calculated the diameter from several different cuts of computed tomogram scans or magnetic resonance imaging. The longest size would be the final diameter. Finally, a retrospective study with a non-randomized design has limited validity. Our study aimed to identify a prognostic role of TBS as well as its impact on survival. To date, there are no well-established criteria focusing on tumor burden classification in head and neck cancer. Given that our study had several limitations inherent to any retrospective study, we for the first time identified that TBS could be used to define the tumor burden for patients with R/M HNSCC and it should have a crucial role in first-line chemotherapy selection.

## 5. Conclusions

Our study was a multicenter retrospective analysis that investigated the impact of PD-L1 and TBS on treatment response for patients with R/M HNSCC. According to TBS, we could easily stratify our R/M HNSCC patients into HTB and LTB groups. Median PFS and OS were significantly worse in patients with HTB as compared with LTB independent of PD-L1 status. Meanwhile, HTB patients treated with EPF had better survival than those treated with PPF, regardless of PD-L1 expression. For LTB PD-L1 positive patients, there was a longer survival with PPF than EPF, while for LTB PD-L1 negative patients, survival was similar between PPF and EPF. In our multivariate analysis, tumor burden was independently associated with PFS and OS. Optimal treatment might be different for R/M HNSCC patients with different tumor burdens. Tumor burden should be taken into consideration in the decision for first-line chemotherapy. Our conclusion is based on real-world evidence and has clinical implications for physicians who treat R/M HNSCC patients. Further prospective randomized control studies are warranted to validate our conclusions.

## Figures and Tables

**Figure 1 cancers-16-01748-f001:**
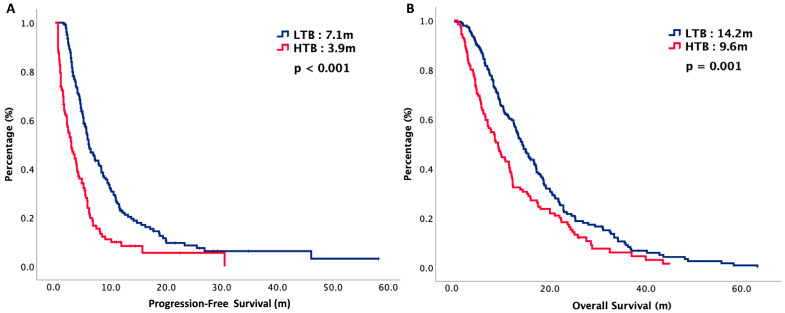
Oncologic outcomes of 252 R/M HNSCC patients, stratified by tumor burden. (**A**) Progression-free survival. (**B**) Overall survival.

**Figure 2 cancers-16-01748-f002:**
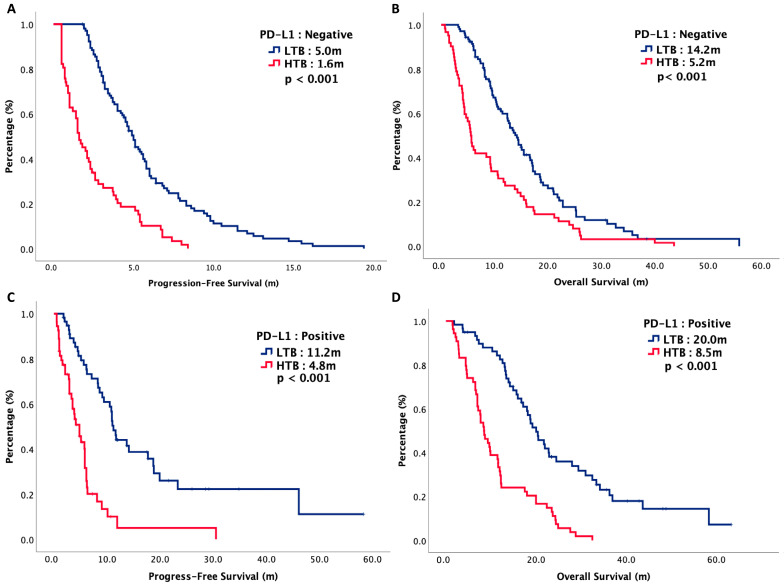
Oncologic outcomes of 252 R/M HNSCC patients, stratified by tumor burden. (**A**) Progression-free survival of PD-L1 negative patients. (**B**) Overall survival of PD-L1 negative patients. (**C**) Progression-free survival of PD-L1 positive patients. (**D**) Overall survival of PD-L1 positive patients.

**Figure 3 cancers-16-01748-f003:**
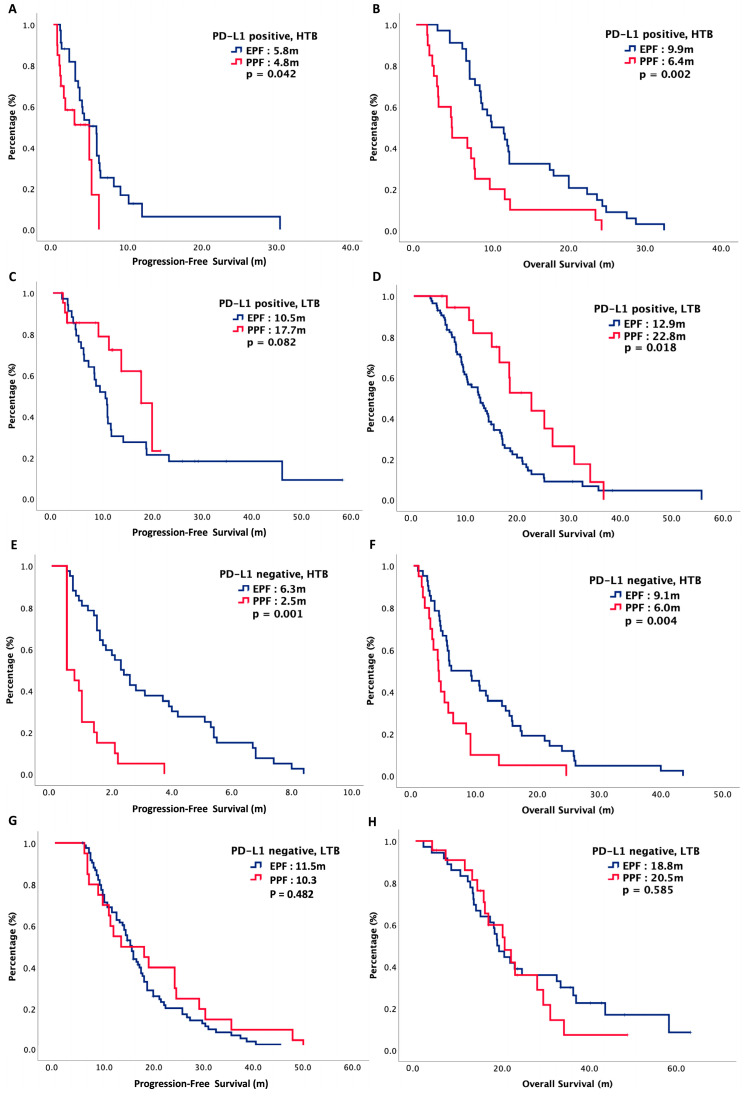
Oncologic outcomes of 252 R/M HNSCC patients, stratified by tumor burden and chemotherapy regimen. (**A**) Progression-free survival of patients with high tumor burden and PD-L1 positive, (**B**) overall survival of patients with high tumor burden and PD-L1 positive, (**C**) progression-free survival of patients with low tumor burden and PD-L1 positive, (**D**) overall survival of patients with high tumor burden and PD-L1 positive, (**E**) progression-free survival of patients with high tumor burden and PD-L1 negative, (**F**) overall survival of patients with high tumor burden and PD-L1 negative, (**G**) progression-free survival of patients with low tumor burden and PD-L1 negative, (**H**) overall survival of patients with low tumor burden and PD-L1 negative.

**Table 1 cancers-16-01748-t001:** Basic characteristics of R/M HNSCC patients, stratified by tumor burden.

	Before PSM					After PSM				
	HTB (*N* = 126)		LTB (*N* = 189)		*p* Value	HTB (*N* = 126)		LTB (*N* = 126)		*p* Value
Gender					0.644					1.000
Male	120	95%	182	96%		120	95%	120	95%	
Female	6	5%	7	4%		6	5%	6	5%	
Age					0.411					0.571
≤60	79	63%	127	67%		79	63%	74	59%	
>60	47	37%	62	33%		47	37%	52	41%	
Primary tumor location					0.600					0.746
Hypopharynx	26	21%	39	21%		26	21%	28	22%	
Oral cavity	60	48%	100	53%		60	48%	64	50%	
Larynx	4	3%	8	4%		4	3%	6	5%	
Oropharynx	36	29%	42	22%		36	29%	28	23%	
P16					0.342					0.961
negative	60	48%	75	40%		60	48%	59	46%	
positive	4	3%	9	5%		4	3%	5	4%	
unknown	62	49%	105	56%		62	49%	62	50%	
Initial T stage					0.010					0.435
T1–T2	36	29%	81	43%		36	29%	42	33%	
T3–T4	90	71%	108	57%		90	71%	84	67%	
Initial N stage					0.106					0.941
N0–N1	51	40%	94	50%		51	40%	50	40%	
N2–N3	75	60%	95	50%		75	60%	76	60%	
Initial M stage					0.076					0.802
M0	109	87%	175	93%		109	87%	110	88%	
M1	17	13%	14	7%		17	13%	16	12%	
Initial stage					0.015					0.912
I–II	15	12%	43	23%		15	12%	16	12%	
III–IV	111	88%	146	77%		111	88%	110	88%	
Curative surgery					0.358					0.797
no	46	37%	58	31%		46	37%	48	38%	
yes	80	63%	131	69%		80	63%	78	62%	
Chemoradiotherapy					0.209					0.908
no	26	21%	53	28%		26	21%	28	22%	
yes	100	79%	136	72%		100	79%	98	78%	
PD-L1 status					0.113					0.409
positive	29	23%	31	16%		29	23%	23	18%	
negative	62	49%	107	57%		62	49%	71	56%	
unknown	35	28%	51	27%		35	28%	32	26%	
Disease status at enrollment					<0.001					0.335
Local recurrence only	29	23%	89	47%		29	23%	36	29%	
Distant metastasis	97	77%	100	53%		97	77%	90	71%	
First-line chemotherapy					0.765					0.247
EPF	86	68%	132	70%		86	68%	77	61%	
PPF	40	32%	57	30%		40	32%	49	39%	

R/M HNSCC, recurrent or metastatic head and neck squamous cell carcinoma; PSM, propensity score match; HTB, high tumor burden; LTB, low tumor burden; PD-L1, programmed death ligand 1; EPF, cetuximab/cisplatin/5-fluorouracil; PPF, pembrolizumab/cisplatin/5-fluorouracil.

**Table 2 cancers-16-01748-t002:** Basic characteristics of 252 R/M HNSCC patients, stratified by tumor burden and treatment.

	HTB (*N* = 126)					LTB (*N* = 126)				
	EPF *N* = 86		PPF *N* = 40		*p* Value	EPF *N* = 77		PPF *N* = 49		*p* Value
Gender					0.325					0.454
Male	83	97%	37	93%		75	97%	45	92%	
Female	3	3%	3	7%		2	3%	4	8%	
Age					0.223					0.369
≤60	57	66%	22	55%		48	62%	26	54%	
>60	29	34%	18	45%		29	38%	23	46%	
Primary tumor location					0.677					0.590
Hypopharynx	18	21%	8	20%		19	25%	9	18%	
Oral cavity	38	44%	22	55%		35	45%	29	59%	
Larynx	3	3%	1	3%		4	5%	2	5%	
Oropharynx	27	31%	9	22%		19	25%	9	18%	
P16					0.126					0.105
negative	46	53%	14	35%		30	39%	29	59%	
positive	3	3%	1	3%		2	3%	3	5%	
unknown	37	43%	25	62%		45	58%	17	36%	
Initial T stage					0.361					0.124
T1–T2	27	31%	9	23%		21	27%	21	43%	
T3–T4	59	69%	31	78%		56	73%	28	57%	
Initial N stage					0.102					0.212
N0–N1	39	45%	12	30%		26	34%	24	49%	
N2–N3	47	55%	28	70%		51	66%	25	51%	
Initial M stage					0.735					0.575
M0	75	87%	34	85%		69	89%	42	85%	
M1	11	13%	6	15%		8	11%	7	15%	
Initial stage					0.426					0.575
I–II	12	14%	3	8%		8	11%	7	15%	
III–IV	74	86%	37	93%		69	89%	42	85%	
Curative surgery					0.478					0.878
no	33	38%	12	30%		29	38%	19	39%	
yes	53	62%	28	70%		48	62%	30	61%	
Chemoradiotherapy					0.559					0.205
no	19	22%	7	18%		14	19%	17	34%	
yes	67	78%	33	82%		63	81%	32	66%	
PD-L1 status					0.907					0.312
positive	20	23%	9	23%		8	10%	15	31%	
negative	42	49%	20	50%		46	60%	25	51%	
unknown	24	28%	11	27%		23	30%	9	18%	
Disease status at enrollment					0.415					0.225
Local recurrence only	18	21%	11	28%		18	23%	18	37%	
Distant metastasis	68	79%	29	72%		59	77%	31	63%	

R/M HNSCC, recurrent or metastatic head and neck squamous cell carcinoma; HTB, high tumor burden; LTB, low tumor burden; PD-L1, programmed death ligand 1; EPF, cetuximab/cisplatin/5-fluorouracil; PPF, pembrolizumab/cisplatin/5-fluorouracil.

**Table 3 cancers-16-01748-t003:** Cox regression analysis of parameters associated with survival.

Variables	PFS		OS	
	HR (95% CI)	*p* Value	HR (95% CI)	*p* Value
Gender, Male vs. Female	0.98 (0.48–2.00),	0.946	0.88 (0.41–1.88)	0.743
Age, ≤60 vs. >60	0.77 (0.56–1.06)	0.109	0.83 (0.60–1.14)	0.240
Primary tumor location, oral cavity vs. others	0.81 (0.61–1.06)	0.128	0.88 (0.65–1.19)	0.398
P16, yes vs. no	0.98 (0.72–1.32)	0.874	0.71 (0.50–1.01)	0.057
Initial T stage, T1–T2 vs. T3–T4	0.95 (0.66–0.38)	0.803	0.73 (0.50–1.06)	0.098
Initial N stage, N0–N1 vs. N2–N3	0.96 (0.68–1.36)	0.812	0.82 (0.56–1.19)	0.289
Initial M stage, M0 vs. M1	0.72 (0.29–1.86)	0.502	0.80 (0.27–2.39)	0.684
Initial stage, stage I–II vs. stage III–IV	0.71 (0.40–1.26)	0.242	0.65 (0.37–1.14)	0.136
Previous radical surgery, yes vs. no	0.63 (0.30–1.33)	0.224	0.62 (0.27–1.42)	0.259
Previous chemoradiotherapy, yes vs. no	0.90 (0.60–1.34)	0.601	0.85 (0.56–1.30)	0.464
Disease status, local recurrence only vs. distant metastasis	0.55 (0.38–0.78)	0.001	0.86 (0.60–1.21)	0.384
PD-L1 expression, negative vs. positive	0.61 (0.44–0.85)	0.004	0.69 (0.48–0.98)	0.039
Tumor burden, LTB vs. HTB	0.23 (0.16–0.34)	<0.001	0.62 (0.44–0.86)	0.005

PFS, progression-free survival; OS, overall survival; HR, hazard ratio; CI, confidence interval; PD-L1, programmed death ligand 1; LTB, low tumor burden; HTB, high tumor burden.

## Data Availability

The datasets that support the findings of this study are available from the corresponding author (M.-C.H.) upon reasonable request.

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
