# Peer review of "Programmed Death Ligand-1 and Tumor Burden Score Dictate Treatment Responses in Patients with Recurrent or Metastatic Head and Neck Squamous Cell Carcinoma"

_cancers, 2024, doi:10.3390/cancers16091748_

Round 1

Reviewer 1 Report

Comments and Suggestions for Authors

This retrospective study sought to evaluate programmed death ligand-1 (PD-L1) and tumor burden score (TBS) as prognostic indicators in patients with recurrent or metastatic HNSCC. The manuscript is interesting due to its original approach to the topic.

 I would have the following observations:

  1. The introduction needs to provide more background and must be improved. It is necessary to explain how pembrolizumab works, especially when it is recommended. Also, tumor burden and PD-L1 should be explained and described in terms of how they can be quantified and correlated. Other recent studies should be cited.
  2. The presentation of the results and discussions must be associated with the data in the tables and figures.
  3. Table 1, Table 2, Figure 2, and Figure 3 are not mentioned in the text.
  4. In the final part of the article, the authors should more clearly highlight the most relevant conclusions of the study and indicate future directions.
  5. It is necessary to access a more extensive bibliography with reference to the information related to Programmed death ligand-1 and tumor burden score or immunotherapy in HNSCC.
  6. Extensive editing of the English language is required.

Overall, the article could be a substantial contribution to the journal. Therefore, I recommend the manuscript for publication after major changes and updates by the authors have been considered.

Comments on the Quality of English Language

  • Extensive editing of the English language is required.

Author Response

Dear editor

Thank you for your consideration about possible publication in Cancers. Here are our responses to reviewers.

Reviewer #1

  1. The introduction needs to provide more background and must be improved. It is necessary to explain how pembrolizumab works, especially when it is recommended. Also, tumor burden and PD-L1 should be explained and described in terms of how they can be quantified and correlated. Other recent studies should be cited.

Response: Thank you for your suggestion. We had added information as your suggestion. As added in Para 1, Line 6-10 of “Introduction”, pembrolizumab is a selective humanized IgG4 kappa monoclonal antibody that inhibits the programmed death-1 (PD-1) receptor. On June 10, 2019, the Food and Drug Administration approved pembrolizumab for the first-line treatment of patients with metastatic or unresectable recurrent head and neck squamous cell carcinoma (HNSCC). The impact of PD-L1 was also provided in introduction. As mentioned in Para 1, Line 10-14 of “Introduction”, the pivotal phase III study Keynote-048 demonstrated that Pembrolizumab monotherapy significantly improved survival in patients with programmed death ligand-1 (PD-L1) combined positive score (CPS) of 1 or more, while pembrolizumab plus platinum, 5-fluorouracil (PPF) extended longer survival in the total population. The impact of tumor burden score was also provided in introduction. As mentioned in Para 2, Line 8-10 of “Introduction”, of note, tumor burden score (TBS) was an indicator of tumor burden and was demonstrated as a novel prognostic factor for colon cancer, hepatocellular carcinoma and cholangiocarcinoma.

2.  The presentation of the results and discussions must be associated with the data in the tables and figures.

Responses: Thank you for your comments. Our results and discussions were all associated with the data in tables and figures. All information was presented in detail.

3. Table 1, Table 2, Figure 2, and Figure 3 are not mentioned in the text.

Responses: Thank you for your review. Table 1 was mentioned in Para 1, line 6-22 of “patients characteristics”. Table 2 was mentioned in Para 2, line 4-7 of “survival outcomes”. Figure 2 was mentioned in Para 1, line 6-12 of “survival outcomes”. Figure 3 was mentioned in Para 2, line 7-21 of “survival outcomes”.

4. In the final part of the article, the authors should more clearly highlight the most relevant conclusions of the study and indicate future directions.

Response: Thank you for your suggestion. As added in Para 1, Line 6-12 of “conclusion”, in our multivariate analysis, tumor burden was an independent predictor that correlated with survival. Optimal treatment might be different for R/M HNSCC patients with different tumor burden. Tumor burden should be taken into consideration in the decision of first-line chemotherapy.

5. It is necessary to access a more extensive bibliography with reference to the information related to Programmed death ligand-1 and tumor burden score or immunotherapy in HNSCC.

Response: Thank you for your suggestion. As mentioned in Para 2, Line 11-12 of “introduction”, Little was known regarding the significance of TBS and PD-L1 for patients with R/M HNSCC. The information related to Programmed death ligand-1 and tumor burden score or immunotherapy in HNSCC is limited. That is the reason why we needed to conduct this study to investigate the correlation between PD-L1, tumor burden score and survival in R/M HNSCC patients.

6. Extensive editing of the English language is required.

Response: Thank you for your suggestion. Our manuscript was revised by a native English speaker.

Reviewer 2 Report

Comments and Suggestions for Authors

The manuscript investigates the prognostic impact of programmed death-ligand 1 (PD-L1) expression and total tumor burden score (TBS) on treatment response in recurrent or metastatic head and neck squamous cell carcinoma (R/M HNSCC) patients. The authors conducted a multicenter retrospective analysis by stratifying patients into high tumor burden (HTB) and low tumor burden (LTB) groups based on TBS. The study finds that patients with LTB generally exhibit better survival outcomes compared to HTB patients, irrespective of PD-L1 status. Additionally, HTB patients treated with cetuximab/cisplatin/5-fluorouracil (EPF) regimen had better survival than those treated with pembrolizumab/cisplatin/5-fluorouracil (PPF), regardless of PD-L1 expression. Conversely, LTB patients with PD-L1 positivity showed longer survival with PPF compared to EPF, while survival outcomes were similar between EPF and PPF for LTB patients with PD-L1 negativity. Multivariate analysis suggests tumor burden as an independent predictor of survival. Overall, the study provides valuable insights into treatment response in R/M HNSCC patients and underscores the importance of considering both tumor burden and PD-L1 status in treatment decision-making. To provide a more comprehensive understanding of the study findings, the study could include the following analysis:

It would be beneficial to provide a more comprehensive overview of the existing literature on R/M HNSCC.

The authors could conduct subgroup analyses based on relevant patient characteristics or tumor status to explore potential interactions and heterogeneity in treatment responses.

Minor:

There are errors with several references: Error! Reference source not found.

Author Response

Reviewer #2

It would be beneficial to provide a more comprehensive overview of the existing literature on R/M HNSCC.

Response: Thank you for your suggestion. We already provided an overview of treatment options of R/M HNSCC patients in “background”. We also provided existing literatures regarding the impact of tumor burden on survival for R/M HNSCC patients in “background”.

The authors could conduct subgroup analyses based on relevant patient characteristics or tumor status to explore potential interactions and heterogeneity in treatment responses.

Response: Thank you for your suggestion. Subgroup analysis to investigate the potential interactions between patients characteristics and survival was conducted with multi-variate analysis, as shown in Table 3.

Minor:

There are errors with several references: Error! Reference source not found.

Response: Thank you for your review. We did not detect any losing information with references. Every reference was cited correctly.

Round 2

Reviewer 1 Report

Comments and Suggestions for Authors

The authors have improved the version of the article following the suggestions made previously. Therefore, the present manuscript form is a publishable work.

Author Response

Dear editor and reviewers

Thank you for your kind review.

Best wishes, 

Mengche, Hsieh